



**A model study of the pollution effects of the first three**
**months of the Holuhraun volcanic fissure**
**B. M. Steensen[1] and M. Schulz[1] and N. Theys[2] and H. Fagerli[1]**
[1]{Norwegian Meteorological Institute, Postbox 43 Blindern, 0313 Oslo, Norway }
[2]{Belgian Institute for Space Aeronomy, Ringlaan-3-Avenue Circulaire, B-1180 Brussels,
Belgium }
Correspondence to: B. M. Steensen (birthems@met.no)
**Abstract**
The volcanic fissure at Holuhraun, Iceland started at the end of August 2014 and continued
for six months to the end of February 2015. Lava floated onto the Holuhraun plain associated
with large $SO_2$ emissions. In this paper we present results from EMEP/MSC-W model
simulations where we added 750 kg/s $SO_2$ emissions at the Holuhraun plain from September
to November. The emission amounted to approximately 4.5 times the daily anthropogenic
$SO_2$ emitted from the 28 European Union countries, Norway, Switzerland and Iceland. Model
results are compared to satellite observations and European surface measurements. The
dispersion but also the ambiguity of the satellite data, due to what is assumed in the retrieval
as a priori $SO_2$ profile, is further explored with model sensitivity runs using different emission
height distributions from the volcano. Satellite-comparable adjusted model vertical column
densities are calculated for the different sensitivity runs where the $SO_2$ mixing ratios from
different vertical layers are weighted with the averaging kernel. The results show the
importance of using the averaging kernel when comparing the model to satellite column
loads, the maximum column densities over 10 DU in the original model data are reduced by
around 50 % due to the weighting. For most days the satellite retrievals have higher mass
burdens values than the adjusted model when summed up over the North Atlantic area. The
discrepancies are explained by the unrealistic constant emission term in the model
simulations, and because the area used for the summation is dependent on the satellite data
detection limit, and the correct position of the model $SO_2$ plume. Surface observations in



Europe showed peak type increases of $SO_2$ concentrations from volcanic plumes passing by
and lasting only for a short time. Three well identified episodes are documented for more
detail. For all the events the timing of the observed concentration peaks compared to the
model quite well. For the first episode presented, the model concentrations are only about
10% to 40% of the observed concentrations. The transport of $SO_2$ to Europe during this event
is found to contribute to very high measured and modelled concentrations at the stations. For
the later plumes, the observed and model concentrations at the stations compare better in
magnitude.  The overall changes in the European $SO_2$ budget due to the volcanic fissure are
estimated. $SO_X$ three monthly wet deposition in the 28 European Union countries, Norway
and Switzerland is found to be more than 30 % higher in the control model simulation with
Holuhraun emission compared to a model simulation with no Holuhraun emission. The
biggest increases, apart from Iceland, are found on the coast of Northern Norway, a region
with frequent precipitation during westerly winds. The total deposition levels in this region
become equal to the most polluted regions over Europe and the average model deposition for
Norway is doubled the level it was back in 1990. For $SO_2$ and $PM_{2.5}$ concentrations, there is
only a ten and six percent increase over Europe between the two model simulations,
respectively. Although the percent increase of $PM_{2.5}$ concentration is highest over
Scandinavia and Scotland, an increase in PM exceedance days is found over Ireland and the
Benelux region. Especially the Benelux region is already very polluted, so that a small
increase in pollution leads to an increase in exceedances days. Although there was a large
increase in total daily emission of $SO_2$ over Europe due to the eruption, Iceland is located too
far away to make a large impact on average pollution levels in the European countries, except
in Iceland itself.
**1   Introduction**
Increased seismic activity in the Bárðarbunga volcano was recorded by the Icelandic Met
Office      from      the      middle      of      August      2014      (http://en.vedur.is/earthquakes-and-
volcanism/volcanic-eruptions/holuhraun/). The activity continued in the volcano but some
tremors appeared also towards the Holuhraun plain, a large lava field north of the Vatnajökull
ice cap, the latter covering the Bárðarbunga and Grimsvötn volcano. On August 31 a
continuous eruption started at Holuhraun with large amounts of lava pouring onto the plain
and large amounts of sulphur dioxide ($SO_2$) emitted into the atmosphere (Sigmundsson et al.



2015). Thordarson and Hartley (2015) estimated $SO_2$ emissions from the magma at
Holuhraun to be around 30 kt/d to 120 kt/d over the first three months of the eruption, with a
maximum during the first two weeks of September. Schmidt et al. (2015) also found that
among several model simulations with different emission fluxes, the model simulations with
the largest emission (120 kt/d) compared best with satellite observations at the beginning of
September. In comparison, Kuenen et al. (2009) estimated the daily anthropogenic emission
from the 28 European Union countries for 2009 to be 13.9 kt/d, while the 2013 estimate is 9.8
kt/d (EMEP, 2015). The eruption ended in February 2015 and during the 6 months of eruption
a total of approximately 11 ($\pm$ 5) Tg $SO_2$ may have been released (Gislason et al. 2015). It is
of interest to investigate the impact of these volcanic emissions on current $SO_2$ levels in
Europe. In the last decades, measures have been taken to reduce $SO_2$ emissions, triggered by
the Convention on Long-range Transboundary Air Pollution (LRTAP), in Europe. Significant
reductions of 75% in emission between 1980 and 2010 are confirmed by observations
(Torseth et al., 2012). The impact of volcanic eruptions with $SO_2$ emissions can thus perturb
the European atmospheric sulphur budget to a larger extent than before and potentially lead to
new acidification of lakes and soils if the eruption would last over a long time period.
For comparison, the big 1783 Icelandic Laki eruption lasted eight months and released a total
amount of estimated 120 Tg of $SO_2$ over eight months. The resulting sulphuric acid caused a
haze observed in many countries of the northern hemisphere and increased mortality in
Northern Europe (Grattan et al., 2003, Thordarson and Self, 2003, Schmidt et al., 2011).  The
fissure at Holuhraun was much weaker than the Laki fissure, both in terms of amount of $SO_2$
released and probably also the height of the eruptive column. Thordarson and Self (1993)
estimated that the Laki erupted at emission heights up to 15 km, while the observations of the
Holuhraun eruptive cloud saw the plume rising up to 4.5 km (vedur.is). Ground level
concentrations exceeded the Icelandic hourly average health limit of 350 µg/m³ over large
parts of Iceland (Gislason et al. 2015).  The World Health Organization (WHO) has a 10
minute limit of 500 µg/m³ and a 24-hour limit of 20 µg/m³. High hourly mean surface
concentrations of $SO_2$ were measured in Ireland (524.2 µg/m³), but then also in Austria (247.0
µg/m³) and Finland (180 µg/m³) (Schmidt et al. 2015, Ialango et al. 2015).
A climate impact of high $SO_2$ emissions may be suspected, such as a cooling of climate due to
an increase in aerosol loadings. Gettelman et al. (2015) using a global climate model found a
small increase in cloud albedo due to the Holuhraun emissions resulting in -0.21 Wm⁻²



difference in radiative flux on the top of the atmosphere.. If the event had happened earlier in
the summer a larger radiative effect could be expected (-7.4 $Wm^{-2}$). Understanding the
atmospheric sulphur budget associated to such events is thus of great interest also for climate
science. The Holuhraun eruption can also serve as a prototype to study ash transport from an
Icelandic volcano. Unlike the two previous big eruptions in Iceland, Eyjafjallajökull in 2010
and Grímsvötn in 2011, this eruption did not emit ash that disrupted air traffic. However,
uncertainties in source estimates, time varying emissions from a point source, dependence of
transport on initial injection height, transport and removal processes from Iceland to Europe
are similar problems for $SO_2$ and ash plumes. Despite Moxnes et al. (2014) showing that $SO_2$
and ash can have different eruption heights, proven capability of modelling the transport of a
$SO_2$ plume can be useful for judging future eruption scenarios where ash can cause a problem.
This study will focus on simulated air quality effects and the perturbed sulphur budget due to
the volcanic $SO_2$ emissions during the first three months, the first two covered also by satellite
observations. Several stations in Europe reported high concentrations of $SO_2$ during this time
and case studies are chosen to evaluate simulated plume development over Europe. The
transport is modelled with the EMEP/MSC-W chemical transport model, one of the important
models used for air quality policy support in Europe during the last 30 years (Simpson et al.
2012). Both station and satellite data are compared to model results to understand the
amplitude and magnitude of the sulphur budget perturbation. A big uncertainty for any
volcanic eruption modelling is the emission term, both with respect to height and magnitude
of the plume. The impact of the height distribution of the emissions is studied through
sensitivity simulations. Finally the perturbed European sulphur budget, as simulated by the
EMEP/MSC-W model, is documented and discussed.
**2   Methods**
**2.1   Model description**
The model simulations of the transport of the $SO_2$ Holuhraun emissions are done with the 3-D
Eulerian chemical transport model developed at the Meteorological Synthesizing Centre-West
(MSC-W) for the European Monitoring and Evaluation Programme (EMEP). The
EMEP/MSC-W model is described in Simpson et al. (2012). Sulphate production from $SO_2$ in
both gas phase and aqueous phase are accounted for. The dry deposition in the model is





parameterized for different land types. Both in-cloud and sub-cloud scavenging are
considered for wet deposition.
The simulations use the EMEP-MACC (Monitoring Atmospheric Composition and Climate)
model configuration. The horizontal resolution of the model simulations is 0.25$^{o}$ (longitude) x
0.125$^{o}$ (latitude). There are 20 vertical layers up to about 100 hPa, with the lowest layer
around 90 meters thick. The model is driven by meteorology from the European Centre of
Medium-Range Weather Forecasts (ECMWF) in the MACC model domain (30$^{o}$ west to 45$^{o}$
east and 30$^{o}$ to 76$^{o}$ north). Iceland is in the upper northwestern corner of the domain, which
implies losses of sulphur from the regional budget terms in sustained southerly and easterly
flow regimes. The meteorology fields used have been accumulated in the course of running
the MACC regional model ensemble forecast of chemical weather over Europe (http://macc-
raq-op.meteo.fr), of which the EMEP/MSC-W model is part of. For our hindcast type
simulations here, only the fields from the first day of each forecast are used. The meteorology
is available with a three hourly interval. All model simulations are run from September
through November 2014.
Emission from the Holuhraun fissure is set to a constant 750 kg/s $SO_2$ (65 kt/d) for the entire
simulation.  For all model runs the anthropogenic emissions are as standard for our EMEP
MACC model configuration. Table 1 shows an overview of the four different model runs that
are used in this study.  For the control run called ctrl_hol, volcanic emissions at Holuhraun are
distributed equally from the ground up to a 3 km emission column height. To test the
sensitivity towards emission height, two additional model simulations are done, low_hol and
high_hol. To derive the impact purely due to the emissions from Holuhraun, a simulation with
no Holuhraun fissure emissions is used, called no_hol.
Anthropogenic $SO_2$ emissions in the model are described in Kuenen et al. (2014). There is a
yearly total $SO_2$ emission of 13.2 Tg/a corresponding to 2009 conditions, the same year that is
used in the reference MACC model configuration. The difference to actual 2014 conditions is
assumed to be unimportant here. The inventory includes 2.34 Tg/a $SO_2$ in yearly ship
emissions over the oceans. Over the continents the yearly emissions are 5.08 Tg/a $SO_2$ for the
28 EU countries, and 5.53 Tg/a $SO_2$ for the non-EU countries in the MACC domain
(including Iceland) covered by the MACC domain.





## 2.2  Observations

The satellite data used in this study stem from the Ozone Monitoring Instrument (OMI) aboard NASA AURA (Levelt et al., 2006). The satellite was launched in July 2004 as part of the A-train earth observing satellite configuration and follows a sun-synchronous polar orbit. The OMI measures backscattered sunlight from the Earth atmosphere with a spectrometer covering UV and visible wavelength ranges. Measurements are therefore only available during daytime. The background $SO_2$ concentrations are often too low to be observable, but increases in $SO_2$ from volcanic eruptions can produce well distinguishable absorption effects (Brenot et al. 2014). Pixel size varies between 13 km x 24 km at nadir and 13 km x 128 km at the edge of the swath. OMI satellite data are affected by "row anomalies" due to a blockage affecting the nadir viewing part of the sensor, which affects particular viewing angles and reduces the data coverage. The zoom-mode of OMI reduces the coverage on some days. The coverage is also reduced by missing daylight, e.g. winter observations from high latitudes are absent. Therefore data from only the two first months from September until the end of October are used in this study.

The retrievals are described in Theys et al. (2015). The sensitivity of backscatter radiation to $SO_2$ molecules varies with altitude (generally decreasing towards the ground level) and therefore the algorithms use an assumed height distribution for estimating the integrated $SO_2$ column density. Since often little information is available at the time of eruption and the retrievals produce results daily (even for days with no eruption) an assumed a priori profile is used for the vertical $SO_2$ distribution. The satellite retrievals used here assume an a priori profile with a plume thickness of 1 km that is centred at 7 km, similar to the method described in Yang et al. (2007). This may be too high for the Bardarbunga eruption, since our simulations indicate that the plume was often situated much lower in the troposphere. Retrieved $SO_2$ column densities may thus be too low. To compare the vertical column density (VCD) from the model to the one from satellite retrievals, the averaging kernel from the satellite has to be used. Each element of an averaging kernel vector defines the relative weight of the true partial column value in a given layer to the retrieved vertical column (Rodgers 2000). Cloud cover also changes the averaging kernel and a spatio-temporally changing kernel is part of the satellite data product (an averaging kernel is provided for each satellite pixel).





To apply the averaging kernel on model data, the satellite data are regridded to the model grid
so that those data from satellite pixels nearest to any given model grid point are used for that
grid point. A smaller area than the whole model domain was chosen to study and compare to
the satellite data, $30^o$ west to $15^o$ east and $45^o$ to $70^o$ north (red boxes in Figure 1). The Aura
satellite does five overpasses over the domain during daytime, swaths are partly overlapping
in the northern regions. For the grid cells where the swaths overlap, the satellite observations
are averaged to produce daily average fields. There are also regions that are not covered by
satellite observation that will not be taken into account in the model data postprocessing. To
make comparable daily averages of the model data, the closest hour in the hourly model
output are matched to the satellite swath time and only grid points that are covered by satellite
are used. The profiles for the averaging kernel in the satellite product are given on 60 levels,
the values from these levels are interpolated to model vertical levels. The new adjusted model
VCD is then calculated by multiplying the interpolated averaging kernel weights to the $SO_2$
concentration in each model layer, integrating all layers with the height of each model layer.
Because of noise in the satellite data small retrieved VCD values are highly uncertain. A
threshold limit is sought to identify those regions that have a significant amount of $SO_2$.
Standard deviation for the satellite data is calculated over an apparently $SO_2$ free North
Atlantic region (size 10 x 15 degrees lat lon respectively), and is found to be around 0.13 DU.
Effects of varying cloud cover are ignored. An instrument detection limit is three times the
standard deviation of a blank, so we assume that with a threshold value set to 0.4 DU we
exclude satellite data below detection limit. Any grid point with a value over this threshold in
the satellite data is used along with the corresponding model data. Daily mass burdens for the
North Atlantic region are calculated by summing up all the $SO_2$ VCD in the grid cells above
the threshold. One DU is $2.69 \ 10^{20}$ molecules per square metre, which corresponds to a
column loading of 28.62 milligrams $SO_2$ per square meter $(mg/m^2)$.
Station data of $SO_2$ surface concentrations are collected by the European Environment
Agency (EEA) through the European Environment Information and Observation Network
(EIONET). We make use of two preliminary subsets of this data, one obtained from work
within the MACC project to produce regular air quality forecasts and reanalysis, and a second
one obtained from EEA as so called up-to-date (UTD) air quality data base, state spring 2015.
The two different subsets cover observation data from different countries, and have not yet
been finally quality assured at the time of writing this paper. We use only station data, which




contain hourly data, however there are missing data and some stations have instruments with
high detection limits making it difficult to create a continuous measurement series with good
statistics. Therefore, in this study some outstanding episodes with high concentrations of $SO_2$
are analysed. Model data are picked consistently from gridded hourly data at model surface
level.
**3   Results**
**3.1   Comparison to satellite data**
Observations by satellite provide information about $SO_2$ location and column density.
Figure 1a shows as an example the VCD from the OMI satellite overpasses on 24 September,
Fig. 1b and Fig. 1c show the modelled and the adjusted VCD from the control run (hol_ctrl).
The observed satellite $SO_2$ cloud and the model simulated $SO_2$ cloud show similar shape and
location. The adjusted model column densities are smaller than the original model VCDs.
More weight is given by the averaging kernel to model layers higher up, close to the reference
height of 7km, where there is less $SO_2$ in our case, with emissions and transport happening in
the lower part of the troposphere. The reduced column densities are more comparable to the
column densities observed by the satellite, there are however some differences of where the
maximum column densities are located.
A quantitative comparison is attempted here by integrating all satellite - and corresponding
model data - above the North Atlantic, between Iceland and Europe, into daily mean column
loads. Figure 2 shows time series from September to October of daily satellite coverage and
daily mass burdens considered over the area where satellite VCD values exceed the 0.4 DU
detection limit as explained above. The area covered by satellite observations at the beginning
of the period is around 70 percent of the domain used here (red boxes in Fig. 1). Towards the
end of the period, the satellite coverage is only around 40 percent because of the increasing
solar zenith angle (a satellite zenith angle cutoff of 75° is used for the satellite data). On some
days, the satellite cover is even lower because of the OMI zoom mode. The percentage of the
satellite data that is above the detection limit is low over the entire two month period, only
reaching around ten percent at the end of September and at the beginning of October.
On most days, the satellite daily mass burdens are above the model value, not including the
days where the zoom mode minimizes the coverage. The average mass burden adjusted to the
7 km reference height for satellite data are 11.17 kt $SO_2$ for satellite and 8.72 kt $SO_2$ for the





model. The highest values are at the beginning of the period, decreasing over time, for both
observed and model mass burdens. Especially during October the values are declining. At the
same time the satellite coverage is decreasing. To further investigate whether the increasing
solar zenith angle is responsible for the increasing bias of the simulated versus observed
VCDs, a new domain further south is used. All that area where satellite observations may be
possible until the end of October (61.25° north) is used to calculate another set of daily
column loads for satellite and model data (see Fig. 2c). Satellite coverage in this southerly
domain is not decreasing over time, but it is also not covering Iceland, so the $SO_2$ from
Holuhraun needs to be transported south to be detected. The plume is transported south four
times over the two-month period as the peaks in column load values show. In this southerly
area the daily accumulated mass burdens are similar in September and in October, supporting
the idea that the decrease in mass burden in Fig. 2b is due to reduced satellite coverage.
Taking into the account the area in which the satellite observed $SO_2$ above detection limit, the
satellite average column loads are calculated as around 70 mg/m$^2$ for the start of the period
and on 19 September, model values are lower. Also the peaks in the middle of October in
Figure 2b have a satellite average column value at 62 mg/m$^2$.
Percentile values from the distribution of the daily mass burden in September and October
2014 from all the three model simulations, original and kernel weighted are shown in Fig. 3.
The kernel weighted model data can be directly compared to the percentile characterisation of
the satellite data. As illustrated in Fig. 1, there is a clear decrease in the column load values
before and after the averaging kernel is applied, because the $SO_2$ plume was found much
below 7 km altitude. The differences between the three model simulations however change
before and after the satellite kernel is applied. For the original model data, the model
simulation with emissions in the lowest kilometre (low_hol) has the highest daily mass
burden values, while the run with the emission highest in the atmosphere (high_hol) exhibits a
lower mass burden than the two other. The higher values in the low_hol simulation can be
explained by less wind and dispersion at low altitudes and thus a more concentrated $SO_2$
cloud than in the two other model simulations. After the averaging kernel is applied to the
model data, the high_hol model simulation has the highest daily values compared to the other
two model simulations. High values in satellite data, and model data with kernel profiles
applied reflect high concentrations and/or volcanic $SO_2$ at high altitudes.



Comparing the satellite data to the kernel weighted model data; the satellite 75[th] percentile is
higher than the model 75[th] percentile. The median for the ctrl_hol, low_hol and high_hol daily
mass burden are 7.38 kt, 4.43 kt and 8.34 kt respectively, for satellite the mass burden median
value is 7.03 kt. The satellite data therefore have higher maximum values that results in the
higher average values and the 75[th] percentile, most of the satellite daily mass burden values
are however around the model data for the ctrl_run. From all the model simulations, with
different emission heights, the ctrl_run is the most similar to the satellite data.
## 3.2  Surface concentrations
$SO_2$ from the volcanic eruption on Holuhraun was measured at several surface stations during
the period. Three different episodes with clear peaks in observed concentrations at stations
around Europe are described in the following paragraphs. Exemplary comparisons are shown
and additional comparisons at other stations are available in the supplementary material.
Figure 4 shows hourly time series for two stations over Great Britain and France from 20
September to 26 September. On 21 September 16 UTC, high $SO_2$ concentrations were
measured at the station in Great Britain. The station is situated in Manchester near the west
coast of Britain. None of the three model simulations exhibits exactly the same values as
observed. Although the model simulations do not reach the observed maximum values, the
model field shows areas south of the station nearby Manchester, where the $SO_2$ concentrations
only due to the volcanic eruption are around 50 $\mu g/m^3$. Interestingly, the agreement of the
model derived volcanic $SO_2$ time series is better in agreement with measurements than the
total simulated $SO_2$ concentration (grey curve), indicating that the model may not resolve
transport from nearby pollution sources and that the station for these days is rather
representative of long range transported $SO_2$. The next day, the plume has moved further
south over France. The French station is situated on the west coast of France in Saint-Nazaire.
The measurements show three peaks over three days, with the highest one measured 12 UTC
23 September. All the three model simulations have the peak concentrations earlier than the
observed, and the concentrations from the model are lower than observed. The three
simulations do however show increased concentrations at the site due to the volcanic eruption
over the three days. The map shows that large parts of France had an increase in $SO_2$ surface
concentrations during this time.





Figure 5 shows the time series for three stations over Scotland and Germany a month later,
from 20 to 26 October. The high_hol simulation shows low concentrations over the Scottish
Grangemouth station, but the ctrl_hol and low_hol have a plume with high concentrations
over the station on 20 October. There are no measurements at this time to compare the model
values to. The timing of the second plume 21 October for the two models is a few hours early
and the modelled concentrations higher than the observed, especially for the low_hol
simulation. The map shows a narrow plume from Iceland south to Scotland and the station
lies on the edge of this plume. On 22 October, the volcanic $SO_2$ is measured at stations in
Germany. Figure 5d shows the plume reaching from Iceland into the North Sea, transported
east and south compared to the situation from the day before. The two stations Kellerwald and
Bremerhaven experience the plume differently. While for Bremerhaven the peak is short the
peak lasts for one day at Kellerwald. The map show that the plume is narrow for all three
stations, and the gradient between where there is no Holuhraun contribution and the
maximum concentration is strong.  At Kellerwald station, the low_hol simulation has the
highest concentrations at the beginning of the plume and the high_hol simulation is highest at
the end of the plume. The ctrl_hol simulation has the most comparable concentrations to the
observed ones, although all the models runs have values that are too high. For the
Bremerhaven station, the observed peak is earlier than the modelled, but all the model runs
match the last hours of the plume.
A third plume is illustrated in Fig. 6 over Northern Europe, occurring from the end of October
to the beginning of November. Figure 6a shows the measured $SO_2$ concentrations at a station
in Oslo, Norway. There are four peaks measured from 29 October to 31, the highest one on 29
October. The models runs show contribution from Holuhraun $SO_2$ over the same three days,
but do not reach the high measured concentrations, especially the first plume is
underestimated. On October 30, the plume is transported south east to Poland. The Polish
station in Sopot experiences a short peak that the model simulates to happen a few hours
earlier. The ctrl_hol simulation has the most comparable concentrations.
Transport to Europe is caused by northerly and north-westerly winds. For the first plume,
where the model shows low concentrations compared to the observations, there had been
southerly winds a time before strong northerly winds transported the $SO_2$ cloud south over
Great Britain and France. Compared to the other two episodes, the $SO_2$ surface concentration
due to Holuhraun are higher over a larger area during this episode. The difficulty of the model
to simulate the SO$_2$ transport correctly depends on the uncertainty in the emission term, the
meteorology fields, the chemical reactions and deposition. Overall the comparison at the
stations and with the satellite data indicates, that the ctrl_hol simulation, with the assumption
that emissions occurred between 0 to 3 km, performs best.

### 3.3   Effects of the eruption on European pollution

The results above show that, although the Holuhraun eruption released large amounts of SO$_2$,
the stations in Europe often measured the increase in SO$_2$ concentrations as short peaks. The
model makes it possible to find a more general view of the impact in the European air quality
due to the volcanic emissions. Table 2 summarizes the model results for Europe. Only grid
cells covering one of the 31 countries are considered when calculating the results shown in
the table, the emission (from anthropogenic sources), concentration and deposition over the
oceans are not studied. Since a large part of the deposition and concentration increase occurs
downwind and close to the emission point, the deposition and concentrations over Iceland is
given in brackets.
The Holuhraun emission estimate used in this study releases over 4.5 times the anthropogenic
emission from the 31 countries (not including ship emissions). The anthropogenic emissions
from Iceland are only 18 kilotons, the SO$_2$ emissions from Iceland increase by over 300 times.
Over the three months, there is 1.32 times more SO$_X$ wet deposition for the control run with
Holuhraun emission than the MACC reference with no Holuhraun emission. The wet
deposition over Iceland and the rest of Europe is dependent on the emission height. The
simulation with the emission highest in the atmosphere (high_hol) has the highest
contribution to the rest of Europe, while less than half of the wet deposition falls on Iceland
compared to the other two runs. For dry deposition, the ten percent increase over Europe is
about the same for all the three model simulations with Holuhraun emissions. For Iceland
however, the SO$_X$ dry deposition is very dependent on emission height.
Figure 7 shows the total deposition over Europe for the standard MACC model simulation
with no Holuhraun emission (no_hol), the control model simulation (ctrl_hol), and the percent
increase for these two model runs. For the no_hol simulation, the highest depositions are over
central and Eastern Europe, while the areas with the lowest deposition are over Iceland,
northern Scandinavia and over the Alps. These are also the areas that experience the highest
percent increase in addition to the northern part of Scotland. Due to the Holuhraun emissions



Iceland has the highest $SO_X$ deposition in Europe, and the coast of northern Norway shows
depositions on the same level as Eastern Europe.
The averaged $SO_2$ surface concentration over Europe is under normal condition higher than
over Iceland. For the simulations with Holuhraun emission the increase over the rest of
Europe is around the same for all three simulations. The ctrl_hol and ctrl_low simulation give
high increases over Iceland, while for the high_hol simulation, the average concentration over
Iceland is close to the rest of Europe.
The increases in $PM_{2.5}$ concentrations are due to increased sulphate production from volcanic
$SO_2$. Dry production is due to $SO_2$ reacting to OH, while wet production occurs in cloud
droplets. $PM_{2.5}$ concentrations are a collection of all aerosol under 2.5 μm, and sulphur is only
a part of the aerosol mass. For $PM_{2.5}$ concentrations, the table shows that Iceland has a lower
average than the rest of Europe for all the four runs, even though Iceland is the contributor to
the increase in pollution levels. The high_hol model simulation has a higher increase in $PM_{2.5}$
concentration than the two other simulations. Especially the low_hol simulations have high
deposition on Iceland, and possibly over the ocean, that will lead to a lower contribution to
the $PM_{2.5}$ increase.
The distribution of $PM_{2.5}$ from the no_hol and ctrl_hol simulation, plotted in Figure 8, shows
the same polluted and clean areas as in Fig. 7. The percent increase is not as high as for the
deposition, but the areas are similar. There is a high increase over north-west Norway and
northern Norway, where the increase is over 100 percent. Figure 8b still shows that although
the percentage increase is high, the $PM_{2.5}$ concentrations in these areas are among the least
polluted in Europe. The high deposition levels in this region indicate that the $PM_{2.5}$ is
scavenged out.
WHO recommends a 24 hourly average mean concentration level of 25 μg/m$^3$ for $PM_{2.5}$ not to
be exceeded over three days over a year (WHO,2005). Figure 9a shows that over the Benelux
region, Northern Germany and Northern Italy this limit value is exceeded by up to ten days
during the three months studied. As the previous plot showed, these are regions with high
average $PM_{2.5}$ concentrations. Because the daily concentrations are already high, any increase
in days in the model ctrl_hol simulation due to the Holuhraun emissions is also occurring in
these regions. The Figure also shows that Northern Ireland experienced up to two exceedance
days due to the volcanic eruption.



**4 Discussion**
The variances between the satellite model data and the satellite observations can be due to
several factors. a) The model emissions flux may be under or overestimated compared to the
real emissions, model VCDs are therefore too low / too large compared to the observed ones.
b) The areas within which the column mass are constructed depend on the threshold VCD
value and the satellite data, so the values in the model depend on the position of the observed
$SO_2$ cloud. If the simulated plume is displaced into an area where the satellite does not show
any useful signal, then this part of the model plume is ignored and may lead to underestimates
of the model. c) The presence of clouds can increase the uncertainty of the satellite retrieval.
d) The unknown real height of the $SO_2$ plumes may introduce additional bias between model
and satellite VCDs.
Our Holuhraun emission term in the three model simulations is constant throughout the
simulations both with respect to emission height and emission flux. Maximum fluxes of 1300
kg/s were reported by Barsotti (2014), and Gislason et al. (2015) estimated a 2.5 times the
average emission term during the first two and a half weeks of the eruption. The assumption
of a constant emission term is thus certainly a simplification. The emission height is also
variable, dependent on initial volcanic eruption characteristics and meteorological conditions
like wind speed and stratification (Oberhuber et al. 1998). A better source estimate for the
eruption is beyond the scope of this study; however the fluctuations in flux magnitude and
emission height can explain some of the differences between observed and simulated
concentrations, especially at the beginning of September.
Ialango et al. (2015) found that the $SO_2$ plume from Holuhraun was detectable with a Brewer
instrument in Finland and compared the measurements to satellite observations from OMI
(Ozone Monitoring Instrument) and OMPS (Ozone Mapping Profiler Suite). From comparing
the ground measured $SO_2$ to the satellite data, the satellite products with an a priori profile
placing the $SO_2$ in the planetary boundary layer gave the best agreement. The reduction in
column loads from applying the averaging kernel seen in this study leading to reasonable
agreement with the satellite VCDs also shows that the $SO_2$ was situated well below the 7 km
altitude. Further comparison of the modelled $SO_2$ vertical distribution to measured one, e.g.
from IASI, is needed to understand the impact of any bias in modelled vertical distribution on
the comparison to satellite derived VCDs. Our sensitivity runs indicate considerable
sensitivity of the estimated amount of $SO_2$ in the North Atlantic area to the vertical



distribution of the $SO_2$. This essentially prevents us from using the satellite data to make a
more quantitative inverse judgement on the emission strength.
Schmidt et al. (2015) presents a comparison between model, satellite and ground observations
for September. Mass burdens from OMI are derived using observed plume heights from the
IASI (Infrared Atmospheric Sounding Interferometer) instrument on the MetOp satellite. The
model NAME (Numerical Atmospheric-dispersion Modelling Environment), a Lagrangian
model, is run for September with sensitivity runs testing both emission height and emission
flux. Comparing with the two satellite data sets, the model simulation with doubled emission
flux (~1400 kg/s) matches well with the OMI satellite data for the first days, while for the rest
of September the model simulation with emission consistent with this paper matches better
(~700 kg/s). The satellite comparison presented here shows that although the satellite data
have higher daily mass burden values for most of the first days, it is not evident that the
emission term on average is too small. The observed plume height presented in Schmidt et al.
(2015) by IASI measurements also supports our ctrl_run emission height distribution between
the ground and 3 km.
Surface concentration comparisons presented in this study and in the supplementary material
show that the volcanic $SO_2$ was observed as short singular peaks lasting from a few hours to
several peaks over a short set of days. The biggest difference for the three studied plumes is
for the first one during September over UK and Western Europe, with up to a factor of four
differences between simulated and measured concentrations at several of the stations. But
both the measured and simulated concentrations during the September event were higher than
the two later events, pointing to a different transport of $SO_2$ in the first event, and not only
higher emissions. Schmidt et al. (2015) also presented a model comparison of observed and
model concentration for these days, and the results show the same as seen in this study. Even
for the NAME model simulation with double emission show smaller concentrations at the
stations presented in Schmidt et al. (2015).
The results in this study show that the sulphur depositions from September to November over
Northern Norway were at the same levels as the most polluted regions in Europe. Emission
ceilings aim set by the Gothenburg Protocol was to reduce the $SO_x$ emissions by 63 % by
2010 compared to the 1990 levels (EMEP, 2015). Most countries have accomplished these
reductions, and the sulphur deposition levels over Europe have decreased. The Holuhraun
eruption changed the picture in some areas. Comparing observed deposition levels at



Tustervatn station in central Norway, the simulated deposition is higher than the yearly
observed averages since 1980. Monthly observed values at this station during the 2011
Grimsvötn eruption show almost as high values as the ctrl_hol simulation. The increase in
$SO_x$ deposition at Birkenes station in Southern Norway is negligible. Northern Norway is
more susceptible for volcanic impact because of the geographical position, in addition to high
frequency of precipitation on the western coast of Norway. Comparing the mean deposition
levels over the three months in 2014 over Norway to model simulations with emissions from
previous years, they are double to the early 1990s (EMEP, 2015). Southern Norway
experienced a sulphur deposition decrease of 40 % from 1980 to 1995 due to emission
abatement in Europe (Berge et al. 1999). The highest contributors to high deposition levels
over Southern Norway were the UK and Germany (18 % and 15 % respectively). Norway
also experienced in 2014 a high percent increase in $PM_{2.5}$ concentrations. The $PM_{2.5}$ levels
over Scandinavia are low, and a small increase in the concentrations leads to high percent
increases. The increase over land shows a similar pattern as the results found in Schmidt et al.
(2011) for a hypothetical Laki eruption. Even though the highest increase is over Scandinavia
and Scotland, the concentrations are too low to exceed the 25 $\mu g/m^2$ limit. Already polluted
regions like the Benelux region experience more days with exceedances as well as North
Ireland.
**5   Conclusions**
The increase in $SO_2$ caused by the volcanic eruption at Holuhraun were observed by satellite
and detected at several stations over Europe. Model simulations with the EMEP/MSC-W
model with emissions from Holuhraun over the period from September to November have
been done to investigate the model capability to simulate such events, and also to study the
impact of the increased emissions on concentrations and depositions over Europe.
The first two months of the model simulations are compared to satellite retrievals from OMI.
The retrievals use an assumed plume height of 7 km. Averaging kernels from the satellite data
are applied on the model data to compare the model data to the satellite. Because of the
weighting, the satellite retrieved mass burden values are dependent on both vertical placement
and amount of $SO_2$. Two sensitivity model simulations with different Holuhraun emission
height are compared to the satellite data together with the control simulation. The results
show the importance of weighting the model data with the averaging kernel when comparing
the model to satellite VCD. The combined uncertainty in emission strength and height impact




when comparing the satellite data to the model simulations makes it difficult to conclude
which emission height is most realistic.
The model simulations are compared to observed concentrations at stations over Europe for
three different events with high concentrations measured at the stations due to the Holuhraun
emissions. For all the events, the timing of the model peaks is well compared to the observed
peaks in concentration. For the two model simulations with emissions distributed lowest in
the atmosphere, a better timing can be seen than for the sensitivity run with the highest
emission height. Due to the transport of $SO_2$ during the first event, both the model data and
measurements are higher than during the two latter events. The biggest difference in
concentration between observed and simulated values is also seen during this first plume
reaching Europe. Uncertainties in the model simulations increase by the length of transport,
and some near misses of the narrow plumes can clearly explain differences between model
and observation. Also, to make a better estimate of the model performance during the whole
volcanic eruption, better quality checked station data is needed.
The change in pollution levels over Europe due to the increase in emissions due to the
volcanic fissure is studied. Of the parameters studied, $SO_X$ wet deposition showed the highest
increase. For the control simulation there is 32 % times more sulphate wet deposition than the
model simulation with no Holuhraun emission over the 28 European Union countries,
Norway and Switzerland. The regions that have the highest increase, apart from Iceland, are
Northern Scandinavia and Scotland, regions that are among the least polluted in Europe.
Especially the coast of Northern Norway, with a percent increase in total deposition of over
1000%, has levels in 2014 equal to the most polluted regions in Europe. Compared to
measurements, the levels are higher than the yearly averaged measured ones at Tustervatn
(Central Norway) since 1980. Compared to model simulations with meteorology and emission
from previous years, the mean deposition levels over Norway are double that of 1990.
The difference in $SO_2$ concentrations over Europe between the no_hol and model simulations
with Holuhraun emission are around 13 percent over the same 30 countries and increases
occurs as short peaks in concentration levels from a few hours to some days. For $PM_{2.5}$
concentration, the increase is six percent. The biggest difference in percent increase are seen
over Scandinavia and Scotland, however these regions are among the cleanest in Europe, also
with the added sulphur caused by the Holuhraun emissions. A lot of the sulphur is also
deposited out over these regions by frequent precipitation. The areas that show increase in





days with over 25 µg/m$^2$ PM$_{2.5}$ concentrations are already polluted. Even though with high
emission from the volcanic fissure at Holuhraun, the increases in pollution levels are low over
Europe.
**Acknowledgements**
Most of the work done for this paper is funded by the Norwegian ash project financed by the
Norwegian Ministry of Transport and Communications and AVINOR. Model and support is
also appreciated trough the Cooperative Programme for Monitoring and Evaluation of the
Long-range Transmission of Air Pollutants in Europe (No: ECE/ENV/2001/003). The
observations are made available trough the EEA UTD database
(http://fme.discomap.eea.europa.eu/fmedatastreaming/AirQuality/AirQualityUTDExport.fmw
) and the MACC project (MACC III project number 633080) obtained with the much
appreciated help of Álvaro Valdebenito. This work has also received support from the
Research Council of Norway (Programme for Supercomputing) trough CPU time granted at
the super computers at NTNU in Trondheim.



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





1    Table 1. Overview of model runs and the Holuhraun emission height assumptions and flux.

| Model run name | Holuhraun layer into which SO2 was injected in the model simulation | Holuhraun flux |
|---|---|---|
| ctrl_hol | 0 - 3 km | 750 kg/s |
| low_hol | 0 - 1 km | 750 kg/s |
| high_hol | 3 - 5 km | 750 kg/s |
| no_hol | | 0 |



Table 2. Emissions, depositions and concentrations for the 28 European Union member states,
Norway and Switzerland for the three months (September, October, November). Emissions
and depositions are total over the three month period, concentrations are the mean over the
period for the 31 countries. Numbers in brackets are the contribution from Iceland, for
emission and deposition, the number represents the sum over Iceland. For concentration, the
number represents the average over Iceland.

| | no_hol | ctrl_hol | low_hol | high_hol | ctrl_hol/no_hol |
|---|---|---|---|---|---|
| Emissions $SO_2$ (kilotons) | 1 257 | 1 257 | 1 257 | 1 257 | 1 |
| | (18) | (5 980) | (5 980) | (5 980) | (5.68) |
| $SO_X$ Wet deposition | 1 043 | 1 382 | 1 285 | 1 465 | 1.32 |
| (kilotons) | (11) | (1 122) | (1 491) | (472) | (2.37) |
| $SO_X$ Dry deposition | 481 | 529 | 524 | 526 | 1.10 |
| (kilotons) | (4) | (151) | (409) | (8) | (1.40) |
| $SO_2$ Surface conc. | 1.39 | 1.58 | 1.56 | 1.56 | 1.13 |
| (mean $\mu g/m^3$) | (0.59) | (38.95) | (105.91) | (1.81) | (66.17) |
| $PM_{2.5}$ Surface conc. | 5.86 | 6.20 | 6.09 | 6.28 | 1.06 |
| (mean $\mu g/m^3$) | (0.82) | (2.50) | (3.13) | (1.12) | (3.06) |



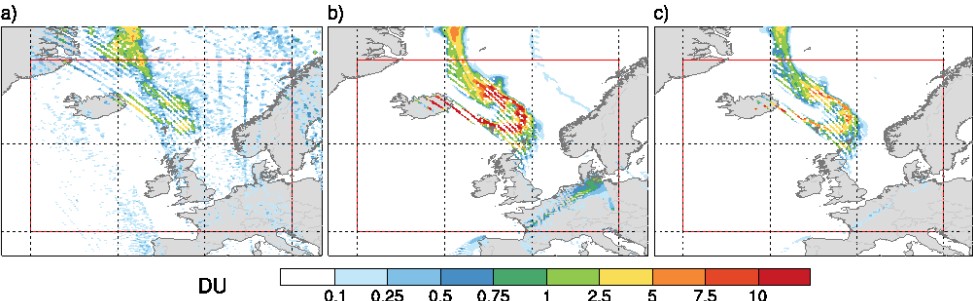

Figure 1. SO$_2$ column density for a) the satellite swaths on 24 September, b) corresponding
model data from 24 September, and c) model data with averaging kernel applied from satellite
data. The red box indicates the area where the satellite statistics in fig.2 are done.

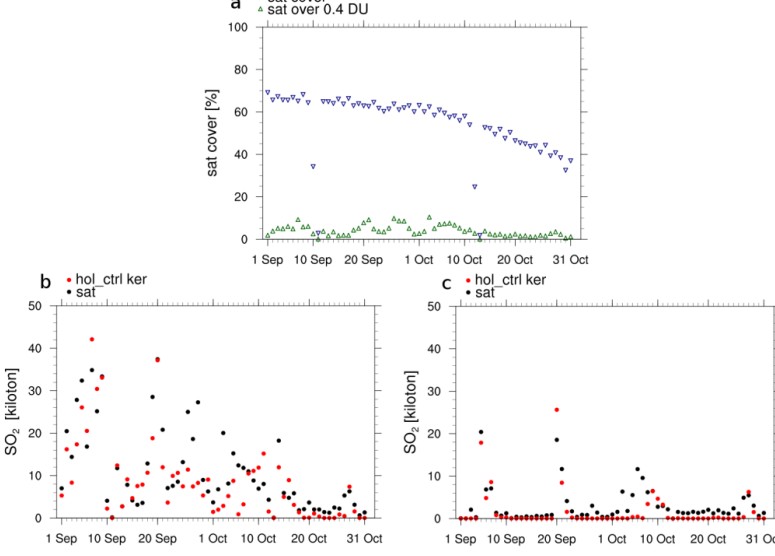

Figure 2. a) Daily time series of satellite observed area coverage (blue triangles) in percent of
the total area of the domain used for the statistics (30 W - 15 E and 45 - 70 N, see fig 1).



Green triangles show the percent of the area where satellite derived $SO_2$ is above 0.4 DU. b)
Daily time series of mass burdens from satellite data (black dots) and from model control run
(red dots) with averaging kernel applied, accumulated in consistent area. c) Shows the same
as b) but over a smaller area just south of 61.15 degrees north.





Figure 3. Distribution of mass burden derived from the 61 daily values  (see fig 2) for the three model simulations, one for each of the three kernel weighted and the satellite data, in the area where satellite derived $SO_2$ exceeds 0.4 DU. The boxes shown represent the 25[th] percentile, the median, and the 75[th] percentile values, lower whiskers the minimum value and upper whiskers the maximum value.

Percentile statistics derived from the 61 daily mass burden values (see fig 2) for the three model simulations, each of the three kernel weighted  and the satellite data, in the area where satellite derived $SO_2$ exceeds 0.4 DU. Boxes show 25[th] percentile, median, and 75[th] percentile values, lower whiskers the minimum value and upper whiskers the maximum value.





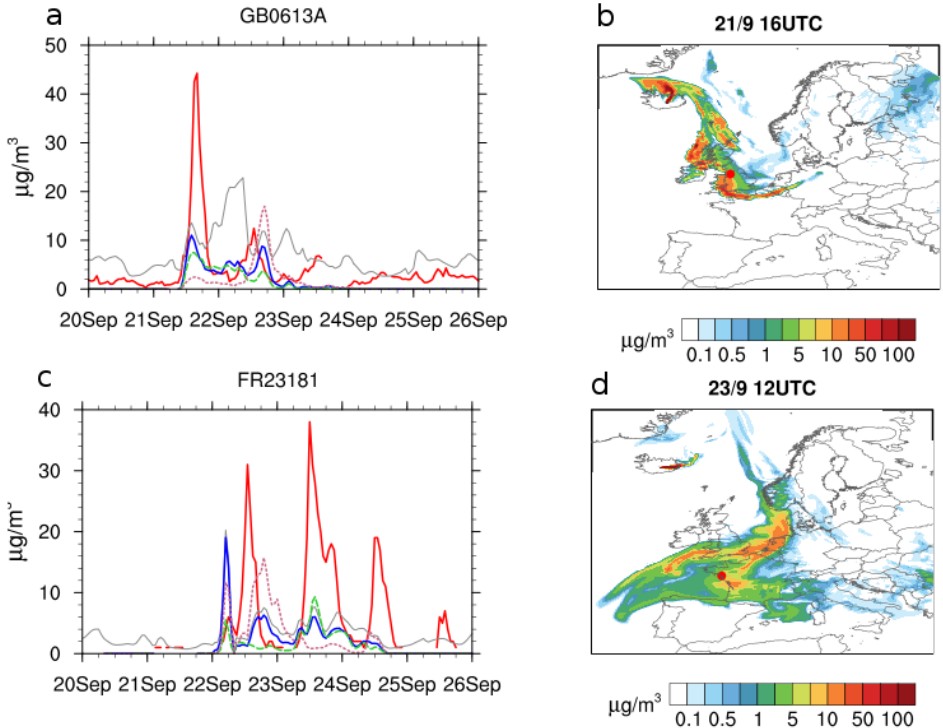

Figure 4. Left: Time Series from 20 to 26 September 2014 for two stations, GB0613A in Manchester and FR23181 in Saint-Nazaire. The red line shows the measured ground concentrations, the grey line represents the modelled ground concentration with ctrl_hol. By subtracting the ground concentrations from the no_hol simulation the concentration due to volcanic eruption for the ctrl_hol, low_hol and high_hol calculated and are shown in the blue, green and pink line respectively. Right: Ground concentration due to the volcanic eruption from ctrl_hol, corresponding to the blue line in the time series, for the time of the maximum observed concentration. The red dot on the map marks the position of the station.



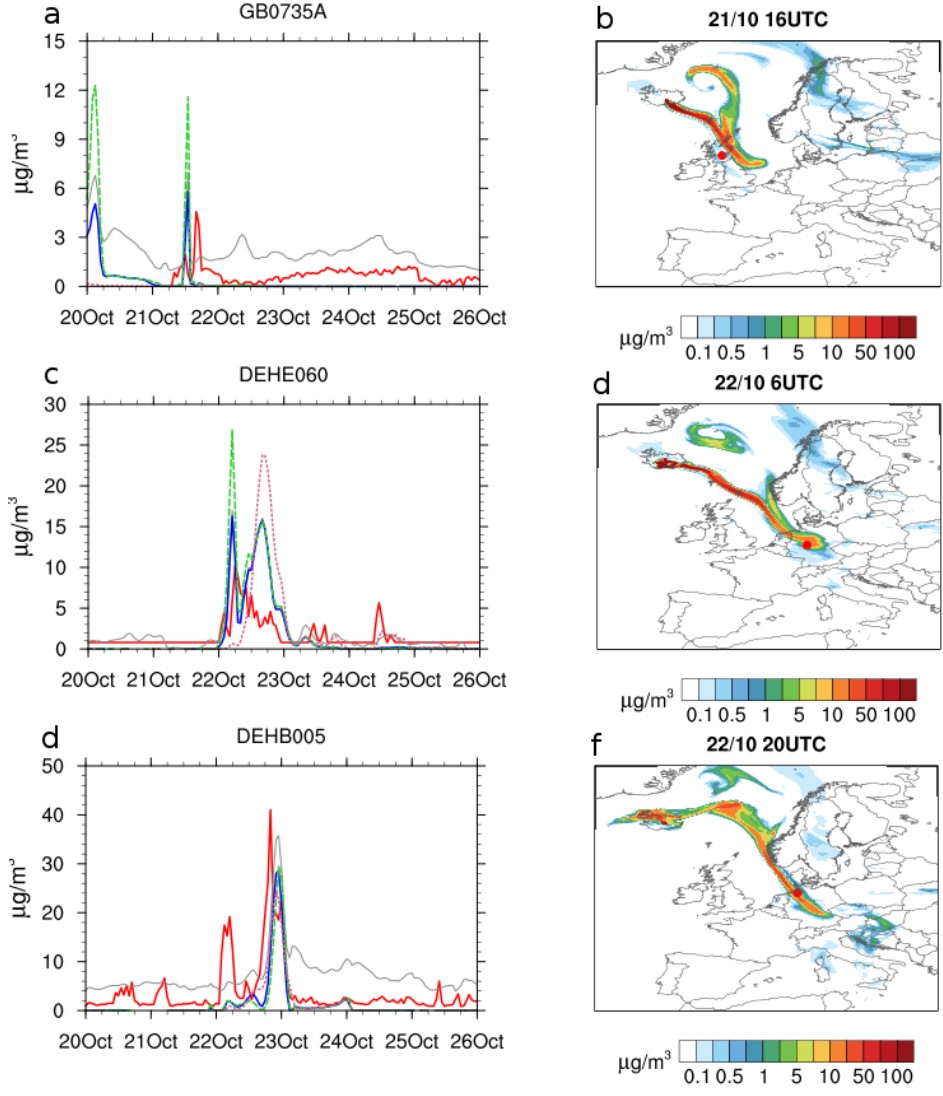

Figure 5. The same as Figure 1, but from 20 to 26 October 2014 for three different stations

GB0735A Grangemouth in Scotland, DEHE060 Kellerwald and DEHB005 Bremerhaven in

Germany.





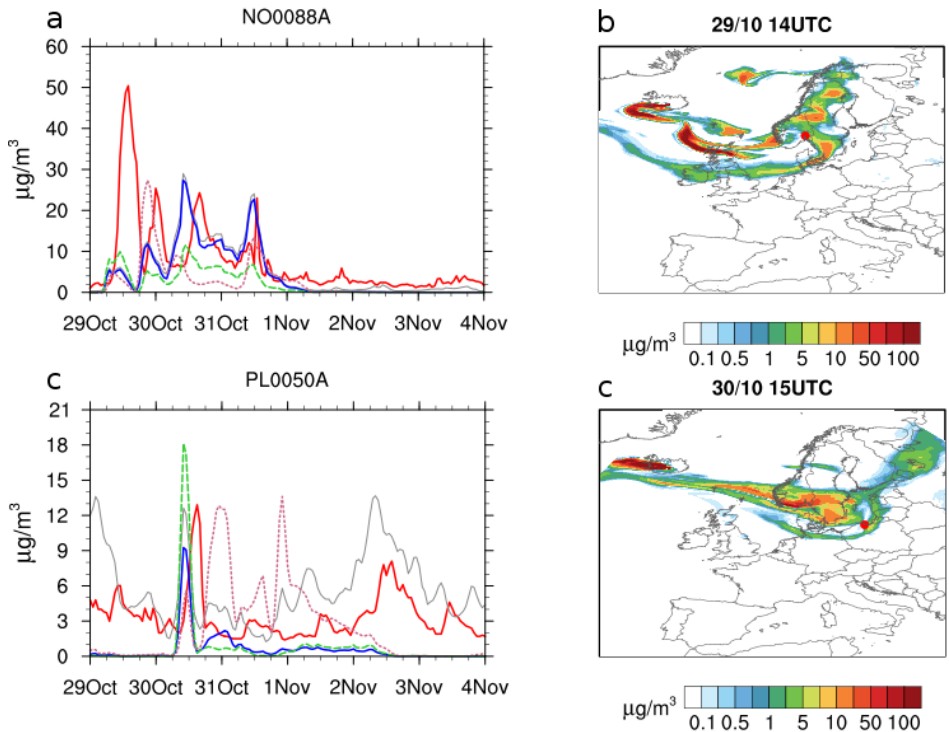

2    Figure 6: The same as the two previous figures but from 29 October to 4 November 2014 for

3    NO0088A Oslo, Norway and PL0050A in Sopot Poland.





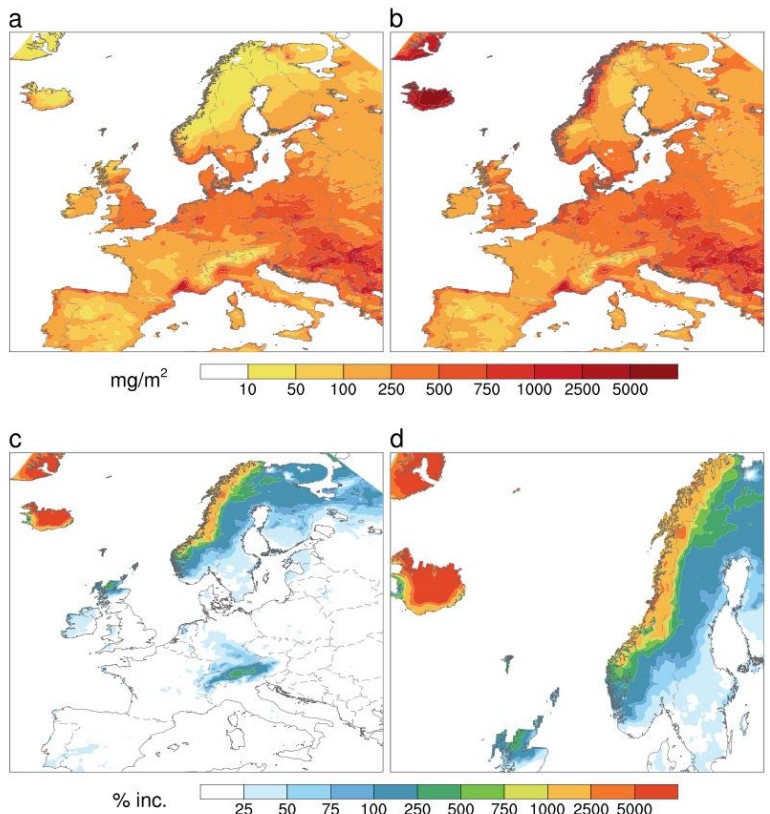

2    Figure 7. Total deposition of $SO_X$ (wet and dry) over Europe from September to November

3    for no_hol (a) and ctr_hol (b) simulations and the percent increase due to the Holuhraun

4    emissions (c). d) Shows the same as c) but zoomed into Norway and Northern Europe.





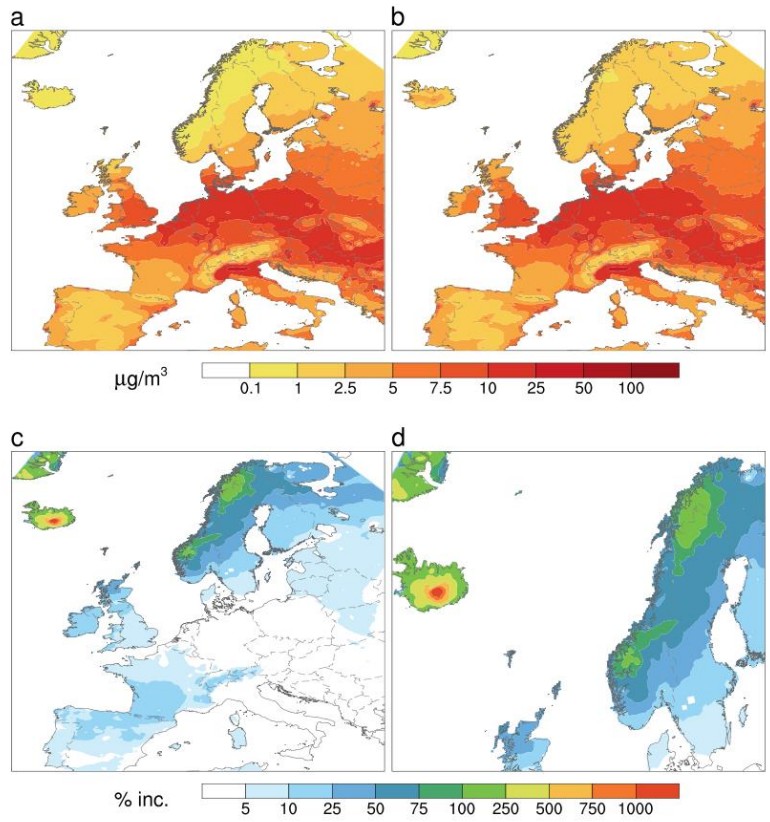

2   Figure 8. Show the same as Figure 7, but with average PM$_{2.5}$ concentration over the three

3   months.

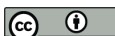



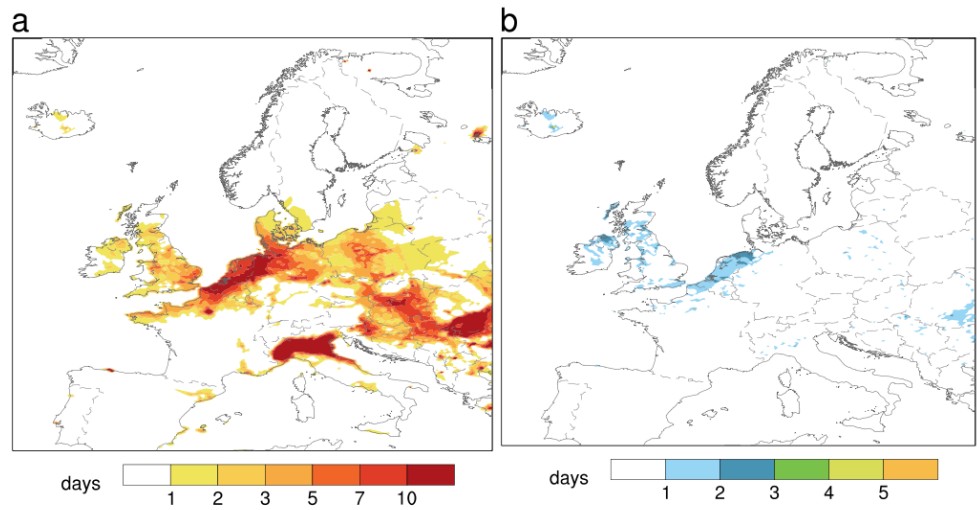

2    Figure 9. a) Days with exceedances of $PM_{2.5}$ over September trough November for the

3    ctrl_hol model simulation. b) The increase in days from no_hol to ctrl_hol.