# Peer review of "A model study of the pollution effects of the first three months of the Holuhraun volcanic fissure: comparison with observations and air pollution effects"

_Atmospheric Chemistry and Physics, 2015_

## Referee Comment (RC1) · Anonymous Referee #2 · 21 Feb 2016

**General comments**

The paper "A model study of the pollution effects of the first three months of the Holuraun volcanic fissure" by Steensen et al. presents an interesting case of a volcanic eruption affecting air quality not only in the vicinity of the eruption area, but also afar, reaching mainland Europe. The paper is in general well written and clear, but it lacks additional in-depth evaluation of the results, specially in relation to the usage of model data to discuss the air quality effects. The text is too descriptive and, even if it states several aspects that may affect the conclusions derived in the study, it does not tackle them nor attempts to describe their potential effects in the air quality extrapolation made in the Results and Discussion sections of the paper.

[Figure]

The authors should address the following general aspects before publication in ACP:

- **Structure and title:** although the title of the paper focuses on the air pollution effects of the Holuhraun fissure eruption, the text is unbalanced in this regard, with a lot of description on the comparison of EMEP simulations with satellite and ground-based measurements. The title should be changed accordingly or the text restructured and reduced. A potential title, matching better the content of the paper, could be "A model study of the three months of the Holuhraun volcanic fissure: comparison with satellite and ground-based data and air pollution effects". The same unbalance exists in the, too long (please reduce), abstract. If the title remains the same, then the paper structure should be modified and the sections on the comparison with ground-based and satellite data should be gathered into a specific section that addresses the performance of the model calculations for this event. The results and discussion should then focus solely on the air pollution aspects once the following item is also addressed.

- **Air pollution effects and chemical transport model results:** the results and discussion on the air pollution effects should be further extended. The text is based solely on one model simulation with evident limitations. More discussion should appear on the potential effects of the mentioned limitations in the overall air quality side of the paper. In addition, the authors present wet and dry deposition results of the simulations with no comparison with existing data. Whenever wet scavenging data exists for such episode, it should be used to assess the very important effect of scavenging. The chemical transport model results are presented and discussed but without the required depth: why are there such large differences in the modelling results and the measurements? Is there a problem in the atmospheric mixing of the EMEP model that leads to such poor representation of the ground base measurements? what are the potential causes of

not only the magnitude differences of the modelled versus measured peaks but also in their times? Have they tested different meteorological fields? Although it is clear that a thorough analysis would probably be out of the scope of the paper, additional thought should be made and added to the manuscript to help the reader with the questions that will surely appear when looking at Figures 4 to 6.

**Specific comments**

Abstract: the abstract is too long and unfocused. Please highlight the main results according to the title of the paper (see General Comments)

Abstract Line 12 - "lava floated" I would change float by flow.

Line 4 Pag. 4 - The authors stated that this case can be used as a proxy for ash events as well. As the authors state further on (lines 9-10) that might not be the case, as Grimvoetn event showed with significantly different transport patterns for SO2 and ash. In addition the processes occurring for ash (including fine and coarse ash, aggregation, gravitational settling...) and SO2 (gas and aqueous phase chemistry) are different enough to add different uncertainties into the processes. It is indeed true that uncertainties in the source term may dominate, but I would rather suggest the authors erase the sentence "The Holuhraun eruption can also serve as a prototype..."

Section 2.1 Model description: it would be useful to the reader to have more information on how the chemical module of EMEP/MSC-W works for SO2 since for this event the reactions with both OH and in the aqueous-phase (due to its low altitude pathway towards Europe) are significant.

Line 12 Pag. 4 - The authors should rewrite this paragraph in order to make it clearer to the reader what are they actually aiming at. What is the MAIN aim? and to achieve such aim what are the SECONDARY milestones or aspects that are addressed?

Line 16 Pag. 5 - Can the authors state (and even better reference) why they are finally using a constant 750 kg/s SO2 flux? They could have easily implemented a variable

emission or taken a "worst case scenario" with the maximum flux of 120kt/day. This affects the discussion on the air pollution section and therefore should be clarified and its implications on the air quality results clearly discussed.

Line 21-23 Pag. 5 - if the authors explain what the control run consists of, also the low and high runs should be explained in addition to the reference of table 1.

Line 4 Pag. 8 - The measurements were regridded? following what method?

Line 10-12 Pag. 12 - It is not entirely clear how the gross numbers in Table 2 are obtained. Is it for the 31 countries but the text states "only grid cells covering ONE ...".

Section 3.3 "Effects of the eruption on European pollution". As stated in the general comments, this section should be extended. In addition, the authors should be careful with too general statements when their conclusions are based solely in one small set of simulations which, from the previous sections, do not prove to be very representative of the concentrations at ground level. Also, please try to add comparisons, whenever possible, with wet deposition measurement data.

---

## Referee Comment (RC2) · Anonymous Referee #1 · 22 Feb 2016

Steensen et al. investigate the effects of the 2014-15 Holuhraun eruption in Iceland on European air quality, SO2 burdens and sulfur deposition using a chemical transport model, satellite data and surface observations of SO2. The study is worth publishing, but in its current form it is not of the scientific standard expected for ACP, mainly because of the very descriptive writing style and lack of detailed comparison to available observations of PM2.5. The abstract should be shortened and throughout the manuscript a much more scientific and quantitative writing style ought to be used. Below I point out some instances that are rather descriptive but this is really a problem throughout most of this manuscript.

Specific comments:

[Figure]

Page 2, line 1: what do you mean by 'peak type' increases? Give numbers here including that date of the measurement and location.

You report increases in PM2.5 mass concentrations based on your model simulations. There are plenty of PM2.5 monitoring sites across Europe (many more than for SO2), so you ought make an effort to compare the model simulations to these observations.

Are there deposition measurements available that could be used to compare to the model simulations?

Page 2, line 31: state the total amount of lava produced

Page 4, line 1: replace 'on the top' with 'at the top'

Page 4, line 4: I strongly disagree with that statement. I agree uncertainties in the source term affect both volcanic gas clouds and ash clouds, but fundamentally the processes that affect SO2 dispersion and conversion to sulfuric acid aerosol particles are different than those that affect volcanic ash concentrations downwind the source. I would simply say that Holuhraun is an eruption worth studying for gas and aerosol processes and effects.

The aims of the study could be described more clearly and put into context with previous studies (e.g. Schmidt et al., 2015, Gislason et al., 2015).

Model description:

It isn't clear to me why the Holuhraun case is called the 'control' simulation. Would it not be more intuitive to call the no_hol simulation the control simulation?

You run sensitivity simulations changing the emission height, but given that your are making statements about effects on air quality, it would be better to also test the sensitivity to the SO2 flux. I would recommend carrying out one simulation using 120 kt/d. It should also be possible to use a time-varying flux by using the data from Thordarson and Hartley (2015) for example.

Observations:

Page 6, lines 23-24: Schmidt et al. (2015) used IASI to derive plume heights, which indicates that using an a priory plume profile of 7 km is too high indeed.

Page 8, lines 3-4: be more specific and state the dates and significance of the SO2 observations for these episodes

Results

3.1 Comparison to satellite data

Page 9, line 1: state the highest value for both the satellite burden and the modeled burdens.

In particular, the simulated burdens for September 2014 should be compared to those in Schmidt et al. (2015), which should give you an opportunity to compare model performance to that of another model.

Page 10, line 7: here you should perform a sensitivity study using higher SO2 emissions than 65 kt/d and discuss the comparison to the satellite-derived burdens.

3.2 Surface concentrations

Page 10, line 9 onwards: give more detailed information including the locations of the measurement stations, the peak values observed and the date/time period of these observations. Surface SO2 mass concentrations of about 500 ug/m3 have been observed in Ireland on 6 September (when the eruption was most powerful). Why do you not use these data as well?

Page 12, lines 3-4: this is only true for the later period of the eruption. You haven't analysed observational data for the early eruption phase, which should be done and it should be stated more clearly that your results support emissions of about 65 kt/d for the late Sep to Oct period.

3.3 Effects of the eruption on European pollution

Page 12, lines 6-7: this has also been shown by Gislason et al. (2015) and Schmidt et al. (2015)

Page 12, line 18 onwards: rewrite all paragraphs using less descriptive writing style

The increases in simulated PM2.5 mass concentrations ought to be compared to measurements from across Europe otherwise the discussion is of little scientific value (in particular because the model is not capturing peak SO2 mass concentrations at the ground compared to the observations).

4 Discussion

First paragraph: several aspects of this discussion are too simplistic because there are observations of the plume height (both at the source and in the far-field using IASI for example)

Second paragraph: Unless you carry out a sensitivity study changing the SO2 flux, you must not state that the variations in the source flux explain the differences between the observations and your model results because you haven't demonstrated that.

Page 15, lines 16-26: state the date and station name for each event that you discuss. I struggle to understand why the difference between the modeled and observed concentrations for the 6 Sep 2014 air pollution event cannot be explained by higher emissions fluxes.

Conclusions

All paragraphs need to be rewritten in a less descriptive manner.

Page 16, line 20: 'increase in SO2' what? Is there a word missing? Do you mean burden or surface mass concentrations? Previous studies that came to the same conclusion should be referenced here.

Last sentence: I disagree; the increase in SO2 mass concentrations was significant in several places even though the pollution episodes were transient.

Figure 1: state which model run is shown.

Figure 3: give date range and how does this compare to Schmidt et al. (2015) who I presume used the same satellite data but state much higher burdens than reported here. Is this down to different averaging periods?
* * *

---

## Editor Comment (EC1) · Y Balkanski (Editor) · 6 Jun 2016

Y Balkanski (Editor)

yves.balkanski@lsce.ipsl.fr

\*\*\*

I think the authors have carefully addressed most of the issues raised by both reviewers. Once the text is cleaned, I think it can be considered ready for publication

\*\*\*\*

---

## Author Response (AR1)

Response to Review #1

We thank the reviewer for taking the time and appreciate the helpful comments and suggestions for improving the manuscript given in this review. We will try to change the writing style to be less descriptive and shorten the abstract.

The comments will be addressed below with review comments stated first, then the author's response in *italic*, the changes to the text is given in quotations (*""*), also in italic.

Specific comments:

Page 2, line 1: what do you mean by 'peak type' increases? Give numbers here including that date of the measurement and location.

*Included in the manuscript, changed peak type to:*

*"Surface observations in Europe showed concentration increases up to 50 μg/m3 averaged over an hour of $SO_2$ from volcanic plumes passing."*

You report increases in PM2.5 mass concentrations based on your model simulations. There are plenty of PM2.5 monitoring sites across Europe (many more than for SO2), so you ought make an effort to compare the model simulations to these observations.

*$PM_{2.5}$ observations are included in the manuscript for the station in Manchester during the first period when both $SO_2$ and $PM_{2.5}$ are measured at the station, for the other $PM_{2.5}$ station with available data over the three periods the plots are in the supplementary data.*

Are there deposition measurements available that could be used to compare to the model simulations?

*When writing the manuscript before submitting to ACPD, these observations were not available. Wet deposition data are now available for some sites, and will be included in the manuscript and supplementary material.*

Page 2, line 31: state the total amount of lava produced

*Included in the manuscript*

Page 4, line 1: replace 'on the top' with 'at the top'

*Changed accordingly*

Page 4, line 4: I strongly disagree with that statement. I agree uncertainties in the source term affect both volcanic gas clouds and ash clouds, but fundamentally the processes that affect SO2 dispersion and conversion to sulfuric acid aerosol particles are different than those that affect volcanic ash concentrations downwind the source. I would simply say that Holuhraun is an eruption worth studying for gas and aerosol processes and effects.

*Removed the sentence and changed the text to:*

*"Unlike the two previous big eruptions in Iceland, Eyjafjallajökull in 2010 and Grímsvötn in 2011, this eruption did not emit ash. However, uncertainties in source estimates, time varying emissions from a point source and dependence of transport on initial injection height are similar problems for $SO_2$ and ash plumes. For eruptions where both ash and $SO_2$ are emitted, $SO_2$ can act as a proxy for ash (Thomas and Prata et al, 2011; Sears et al., 2013), however separation can occur both because of*

*different eruption heights within the plume  (Moxnes et al., 2014) and  density differences after some time. Proven capability of modelling the transport of a volcanic plume can be useful for judging future eruption scenarios where ash may cause a problem."*

The aims of the study could be described more clearly and put into context with previous studies (e.g. Schmidt et al., 2015, Gislason et al., 2015).

*The aim is to study the perturbed sulphur budget due to the volcanic emission, both observed and modelled. The second aim is investigate the impact of the eruption on European pollution levels. This is also made more clear in the manuscript.*

Model description:

It isn't clear to me why the Holuhraun case is called the 'control' simulation. Would it not be more intuitive to call the no_hol simulation the control simulation?

*The control simulation is renamed basic (bas). From the observed heights, and emission fluxes given elsewhere, this simulation is the "best guess" simulation.*

You run sensitivity simulations changing the emission height, but given that your are making statements about effects on air quality, it would be better to also test the sensitivity to the SO2 flux. I would recommend carrying out one simulation using 120 kt/d. It should also be possible to use a time-varying flux by using the data from Thordarson and Hartley (2015) for example.

*Increase in the $SO_2$ flux will lead to higher numbers, however the increase is close to linear to the increase in emission flux. This is also shown in the paper by Schmidt et al. (2015)  and in Figure 1, where a sensitivity simulation with 120 kt/d emission (called max volc), and  a simulation with the time varying Thordarson and Hartley (2015) emission is  plotted (Thor volc). However comparing this simulation with the satellite data show worse result. This indicates that the height of the emission is important, and the transportation towards the station.*

[Figure]

*Figure 1. Measured and modelled comcentration at GB0613A station in Manchester, Great Brittain (red dots on the map). The timeseries above show $SO_2$ concentrations and below for $PM_{2.5}$ for observed (red) and five different model simulations, bas all show all sources for $SO_2$ and $PM_{2.5}$, while the other volc lines only show values due to the volcanic eruption. The time of the map plot is at the maximum observed concentrations.*

Observations:

Page 6, lines 23-24: Schmidt et al. (2015) used IASI to derive plume heights, which indicates that using an a priory plume profile of 7 km is too high indeed.

*Changed in the manuscript to:*

*"As found in Schmidt et al. (2015), this is too high for the Bardarbunga eruption therefore retrieved $SO_2$ column densities may thus be too low"*

Page 8, lines 3-4: be more specific and state the dates and significance of the SO2 observations for these episodes

*Included in the text:*

*"For the first six day period, between 20 to 26 September, high concentrations of $SO_2$ were measured over Great Brittain, and countries to the south. For the second six day period, a month later (20 to 26 October) the plume was also detected over Great Brittain, but transported further east towards Germany. For the last plume studied here from 29 October to 4 November, the volcanic emission was transported southeast to the coast of Norway and countries to the south. Model data to represent the station values are picked from hourly data at model surface level in the grid cell where the station is located."*

Results

3.1 Comparison to satellite data

Page 9, line 1: state the highest value for both the satellite burden and the modeled burdens.

*Added in the manuscript.*

*"The highest values are at the beginning of the period, 42.11 kt $SO_2$ for the model data on 7 September, and 37.42 kt $SO_2$ 20 September for the satellite data."*

In particular, the simulated burdens for September 2014 should be compared to those in Schmidt et al. (2015), which should give you an opportunity to compare model performance to that of another model.

*The simulated burdens presented in this study and the simulated burdens presented in Schmidt et al. (2015) Figure 4 are not directly comparable. The model burdens are weighted with the kernel to compare to the satellite data while in Schmidt et al. (2015), the a priori satellite height in the OMI data are set to the observed heights by IASI to compare to the NAME model results with emission heights at 1.5 to 3 km. Both plots show however higher satellite burdens compared to model on 4 September and higher model burdens compared to satellite on 6 and 7 September. This is included in the discussion part of the manuscript.*

Page 10, line 7: here you should perform a sensitivity study using higher SO2 emissions than 65 kt/d and discuss the comparison to the satellite-derived burdens.

*Figure 2 show the same as Fig. 2 b in the manuscript, but with the time varying emission term from Thordarson and Hartley (2015). The $SO_2$ is released in the same height as for the basic model run, between 0 and 3 km. Although matching better for the first days, the results are not better overall. All the results presented in the manuscript show that the dependency of emission height is more important. This is included in the discussion part of the manuscript.*

[Figure]

*Figure 2. Daily time series of mass burdens from satellite data (black dots) and from model run with Thorarson and Hartley (2015) emission (red dots) with averaging kernel applied.*

3.2 Surface concentrations

Page 10, line 9 onwards: give more detailed information including the locations of the measurement stations, the peak values observed and the date/time period of these observations. Surface SO2 mass concentrations of about 500 ug/m3 have been observed in Ireland on 6 September (when the eruption was most powerful). Why do you not use these data as well?

*The detailed information will be included in the manuscript. The high SO$_2$ concentration observed over Ireland on 6 September did not show up on many of the station that we were able to collect, so it was left out of the manuscript, but two Irish stations are shown in Figure 3.*

[Figure]

*Figure 3. Measured and modelled comcentration at station IE0028A and IE0108A in Ireland (red dots on the map). The timeseries show SO$_2$ concentrations and for observed (red) and five different model simulations, bas all show all sources for SO$_2$, while the other volc lines only show values due to the volcanic eruption. The time of the map plot is at the maximum observed concentrations.*

*Station IE0028A lies east of station IE0108A where the observed concentrations are higher. The concentration maps also show that concentrations over 100 μg/m3 over the North Atlantic Ocean to the west of Iceland in an anticyclone. Both the stations have higher concentrations for the simulation where the emissions is put between 3 to 5 km. Schmidt et al (2015) found the same result, and analyzed the discrepancies to be a problem with the boundary layer height. The satellite comparison for this time shows that the model data have higher values than the satellite observations.*

Page 12, lines 3-4: this is only true for the later period of the eruption. You haven't analysed observational data for the early eruption phase, which should be done and it should be stated more clearly that your results support emissions of about 65 kt/d for the late Sep to Oct period.

*The satellite data comparison does not clearly show that the column burdens are too low at the beginning of the period, apart from the first few days, but on September 6, the model has higher summed SO₂ value than the observed satellite over the larger area (not the smaller). Both Figure 1 and Schmidt et al. (2015) found that the model runs with the higher emission altitude have higher concentrations at the sites, the satellite time series of this model simulations show that the model data have even higher values (Figure 4). These results points in two different directions, and it is difficult to conclude that the emission flux should be higher and at a higher level although Schmidt et al. (2015) found this. The higher concentrations in the observations seem to come from the boundary layer being badly represented in the meteorological data. This is included in the discussion part of the manuscript.*

[Figure]

*Figure 4. Daily time series of mass burdens from satellite data (black dots) and from model run with emissions released between three and five km, high_hol (red dots) with averaging kernel applied.*

*To maybe clarify more, the text is changed to:*

*"Overall the comparison to observations, both satellite and station data, the bas_hol model simulation match best with the observed satellite column burdens and with the timing and for some stations concentrations of the observed peaks."*

3.3 Effects of the eruption on European pollution

Page 12, lines 6-7: this has also been shown by Gislason et al. (2015) and Schmidt et al. (2015)

*Added the references.*

Page 12, line 18 onwards: rewrite all paragraphs using less descriptive writing style

*Will change the writing style*

The increases in simulated PM$_{2.5}$ mass concentrations ought to be compared to measurements from across Europe otherwise the discussion is of little scientific value (in particular because the model is not capturing peak SO2 mass concentrations at the ground compared to the observations).

*PM$_{2.5}$ is included in the station comparisons, where a station both measures PM$_{2.5}$ and SO$_2$ in the paper and the other stations in supplementary material.*

Discussion

First paragraph: several aspects of this discussion are too simplistic because there are observations of the plume height (both at the source and in the far-field using IASI for example)

*Although presenting plume heights, Schmidt at al. (2015) does not use these heights for their model simulations, and there are some discrepancies in the calculations especially the am data on 15 September where the center of mass is 4 km and the plume height is only 3.9 km. The authors agree that the height is not unknown so included it in the discussion.*

*Changed the text to:*

*"d) Schmidt et al. (2015) presents IASI (Infrared Atmospheric Sounding Interferometer) plume heights between 5.5 km to 1.6 km derived from an area of 500 km around the volcanic location, and a mean IASI centre of mass height between 2.7 km to 0.6 km. The fluctuating real height of the $SO_2$ plume may introduce additional bias between model and satellite VCDs."*

Second paragraph: Unless you carry out a sensitivity study changing the SO2 flux, you must not state that the variations in the source flux explain the differences between the observations and your model results because you haven't demonstrated that.

*Model runs with different emission fluxes are presented in the answer here. The almost linear increase of concentrations with emission is also presented in Schmidt et al. (2015). Variations in emission flux can also change within an hour, so unless a more thoroughly study is done for the emission term, this is an uncertainty factor.*

Page 15, lines 16-26: state the date and station name for each event that you discuss. I struggle to understand why the difference between the modeled and observed concentrations for the 6 Sep 2014 air pollution event cannot be explained by higher emissions fluxes.

*Added the information in the text.*

*For the 6 September event, as discussed above, the satellite results and the concentrations at the stations show discrepancies in terms of concentrations, other studies points towards a higher emission during this first week. The models (both EMEP and NAME) fail to simulate the high concentrations even with higher emissions. Schmidt et al. (2015) points towards the model not being able to reproduce the atmospheric subsidence and the representation of the boundary layer from the meteorological field.*

Conclusions

All paragraphs need to be rewritten in a less descriptive manner.

*Will change the writing style*

Page 16, line 20: 'increase in SO2' what? Is there a word missing? Do you mean burden or surface mass concentrations? Previous studies that came to the same conclusion should be referenced here.

Changed the sentence to:

*The increase in emitted $SO_2$ to the atmosphere caused by the volcanic eruption at Holuhraun were observed by satellite and detected at several stations over Europe (Schmidt et al. 2015).*

Last sentence: I disagree; the increase in SO2 mass concentrations was significant in several places even though the pollution episodes were transient.

*Changed it to:*

*"Even with high emissions from the volcanic fissure at Holuhraun, the increase in pollution levels over Europe is low, with only transient episodes with high increases in $SO_2$ concentration."*

Figure 1: state which model run is shown.

*Added to the caption.*

Figure 3: give date range and how does this compare to Schmidt et al. (2015) who I presume used the same satellite data but state much higher burdens than reported here. Is this down to different averaging periods?

*This is explained above. The a priori height used by the retrieval of OMI satellite is different.*

Response to Review #2

We thank the reviewer for taking the time and appreciate the helpful comments and suggestions for improving the manuscript given in this review.

The comments will be addressed below with review comments stated first, then the author's response in *italics*, the changes to the text is given in quotations (*""*), also in italics.

**Structure and title:** although the title of the paper focuses on the air pollution effects of the Holuhraun fissure eruption, the text is unbalanced in this regard, with a lot of description on the comparison of EMEP simulations with satellite and ground-based measurements. The title should be changed accordingly or the text restructured and reduced. A potential title, matching better the content of the paper, could be "A model study of the three months of the Holuhraun volcanic fissure: comparison with satellite and ground-based data and air pollution effects". The same unbalance exists in the, too long (please reduce), abstract. If the title remains the same, then the paper structure should be modified and the sections on the comparison with ground-based and satellite data should be gathered into a specific section that addresses the performance of the model calculations for this event. The results and discussion should then focus solely on the air pollution aspects once the following item is also addressed.

*The abstract will be shortened, and the authors agree that the title does not reflect the context of the manuscript, title is changed to:*

*"A model study of the pollution effects of the first three months of the Holuhraun volcanic fissure: comparison with observations and air pollution effects"*

**Air pollution effects and chemical transport model results:** the results and discussion on the air pollution effects should be further extended. The text is based solely on one model simulation with evident limitations. More discussion should appear on the potential effects of the mentioned limitations in the overall air quality side of the paper. In addition, the authors present wet and dry deposition results of the simulations with no comparison with existing data. Whenever wet scavenging data exists for such episode, it should be used to assess the very important effect of scavenging. The chemical transport model results are presented and discussed but without the required depth: why are there such large differences in the modelling results and the measurements? Is there a problem in the atmospheric mixing of the EMEP model that leads to such poor representation of the ground base measurements? what are the potential causes of not only the magnitude differences of the modelled versus measured peaks but also in their times? Have they tested different meteorological fields? Although it is clear that a thorough analysis would probably be out of the scope of the paper, additional thought should be made and added to the manuscript to help the reader with the questions that will surely appear when looking at Figures 4 to 6.

*More on the limitations of the model for not performing better for the high concentration events will be included in the discussion part. Comparison to $PM_{2.5}$ measurements and $SO_x$ wet deposition measurements will be included for stations where it is available. There is no known problem in the atmospheric mixing in the EMEP model. The complex transport to the stations for the first episode with first southerly winds, then northerly caused the $SO_2$ to stay in the atmosphere longer and increase in concentrations. The uncertainties due to model representations and meteorology errors accumulate and create the discrepancies seen in Figure 4. The comparison is better for the two later periods. The ECMWF meteorology is the best available meteorology for the EMEP model, and the resolution is also high. Schmidt et al. (2015) use another meteorological driver and also find the same*

*discrepancies over this late September period. The result and discussion part will be extended to include more station comparison data.*

Specific comments

Abstract: the abstract is too long and unfocused. Please highlight the main results according to the title of the paper (see General Comments)

*The abstract will be shortened and more focused.*

Abstract Line 12 - "lava floated" I would change float by flow.

*Changed accordingly.*

Line 4 Pag. 4 - The authors stated that this case can be used as a proxy for ash events as well. As the authors state further on (lines 9-10) that might not be the case, as Grimvoetn event showed with significantly different transport patterns for SO2 and ash. In addition the processes occurring for ash (including fine and coarse ash, aggregation, gravitational settling...) and SO2 (gas and aqueous phase chemistry) are different enough to add different uncertainties into the processes. It is indeed true that uncertainties in the source term may dominate, but I would rather suggest the authors erase the sentence "The Holuhraun eruption can also serve as a prototype..."

*The authors agree that $SO_2$ is not a prototype for ash, removed the statement and changed the text to:*

*"Unlike the two previous big eruptions in Iceland, Eyjafjallajökull in 2010 and Grímsvötn in 2011, this eruption did not emit ash. However, uncertainties in source estimates, time varying emissions from a point source and dependence of transport on initial injection height are similar problems for $SO_2$ and ash plumes. For eruptions where both ash and $SO_2$ are emitted, $SO_2$ can act as a proxy for ash (Thomas and Prata et al, 2011; Sears et al., 2013), however separation can occur both because of different eruption heights within the plume (Moxnes et al., 2014) and density differences after some time. Proven capability of modelling the transport of a volcanic plume can be useful for judging future eruption scenarios where ash may cause a problem."*

Section 2.1 Model description: it would be useful to the reader to have more information on how the chemical module of EMEP/MSC-W works for SO2 since for this event the reactions with both OH and in the aqueous-phase (due to its low altitude pathway towards Europe) are significant.

*Extended the model description to:*

*"$SO_2$ is oxidized to sulphate in both gas and aqueous phase with assumed equilibrium. In gas phase the oxidation is initiated by Hydroxide (OH), OH is labelled "short lived" and is controlled by local chemistry. In aqueous phase the oxidants ozone, hydrogen peroxide and oxygen catalysed by metal ions contribute to oxidation."*

Line 12 Pag. 4 - The authors should rewrite this paragraph in order to make it clearer to the reader what are they actually aiming at. What is the MAIN aim? and to achieve such aim what are the SECONDARY milestones or aspects that are addressed?

*The aim is to study the perturbed sulphur budget due to the volcanic emission, both observed and modelled. The second aim is investigate the impact of the eruption on European pollution levels. This is also made more clear in the manuscript.*

Line 16 Pag. 5 - Can the authors state (and even better reference) why they are finally using a constant 750 kg/s SO2 flux? They could have easily implemented a variable emission or taken a "worst case scenario" with the maximum flux of 120kt/day. This affects the discussion on the air pollution section and therefore should be clarified and its implications on the air quality results clearly discussed.

*A worst case scenario with a emission of 1400 kg/s (max_hol) and a time varying emission given in Thordarson and Hartley (2015)(Thor_hol) is also studied, but the results were not better compared to observations. As shown in the Figure , 1for concentration comparison at the Manchester station in September where it is shown that an increase in emission gives an almost linear increase in concentrations of $SO_2$ and $PM_{2.5}$ (and deposition, not shown). Figure 2 show the satellite comparison for the hol_Thor simulation, same as Figure 2b in the manuscript.*

[Figure]

*Figure 1. measured and modelled comcentration at GB0613A station in Manchester, Great Brittain (red dots on the map). The timeseries above show $SO_2$ concentrations and below for $PM_{2.5}$ for observed (red) and five different model simulations, bas all show all sources for $SO_2$ and $PM_{2.5}$, while the other volc lines only show values due to the volcanic eruption. The time of the map plot is  the time of maximum oberved concentration.*

[Figure]

*Figure 2. Daily time series of mass burdens from satellite data (black dots) and from model run with Thorarson and Hartley (2015) emission (red dots) with averaging kernel applied.*

*The height of the emission is seen to be more important, and therefore these two simulations are not included in the manuscript. This discussion is included in the manuscript, and the number behind the emission is added in the text:*

*"Emission from the Holuhraun fissure is set to a constant 750 kg/s $SO_2$ (65 kt/d) for the entire simulation from the total $2.0 \pm 0.6$ Tg $SO_2$ emitted in September estimated in Schmidt et al. (2015). "*

Line 21-23 Pag. 5 - if the authors explain what the control run consists of, also the low and high runs should be explained in addition to the reference of table 1.

*Will include more description.*

Line 4 Pag. 8 - The measurements were regridded? following what method?

*The sentence is changed to:*

*"Model data to represent the station values are picked from hourly data at model surface level in the gridpoint where the station is located."*

Line 10-12 Pag. 12 - It is not entirely clear how the gross numbers in Table 2 are obtained. Is it for the 31 countries but the text states "only grid cells covering ONE ...".

*Thank you for pointing out that this it is not clear. The sentence is changed to:*

*"Grid cells covered by the countries mentioned are used for calculating the results shown in the table,"*

Section 3.3 "Effects of the eruption on European pollution". As stated in the general comments, this section should be extended. In addition, the authors should be careful with too general statements when their conclusions are based solely in one small set of simulations which, from the previous sections, do not prove to be very representative of the concentrations at ground level. Also, please try to add comparisons, whenever possible, with wet deposition measurement data.

*The section will be extended to include more comparison to the station data observations, and rewritten so the statements better reflect the uncertainty that comes from a single model study.*

**A model study of the pollution effects of the first three months of the Holuhraun volcanic fissure: comparison with observations and air pollution effects**

**B. M. Steensen[1] and M. Schulz[1] and N. Theys[2] and H. Fagerli[1]**

[1]{Norwegian Meteorological Institute, Postbox 43 Blindern, 0313 Oslo, Norway }

[2]{Belgian Institute for Space Aeronomy, Ringlaan-3-Avenue Circulaire, B-1180 Brussels, Belgium }

Correspondence to: B. M. Steensen (birthems@met.no)

**Abstract**

The volcanic fissure at Holuhraun, Iceland started at the end of August 2014 and continued for six months to the end of February 2015. Lava flow onto the Holuhraun plain combined with  $SO_2$ emissions amounting up to approximately 4.5 times the daily anthropogenic $SO_2$ emitted from the 28 European Union countries, Norway, Switzerland and Iceland. In this paper we present results from EMEP/MSC-W model simulations where we added 750 kg/s $SO_2$ emissions at the Holuhraun plain from September to November.  at three different emission heights. Model results are compared to satellite observations and European surface measurements. The different  runs are weighted with the satellite averaging kernel, the effect of the weighting

 are dependent on the height of the sulphur dioxide in the atmosphere. Surface observations in Europe showed  concentration increases up to 50 µg/m3 averaged over an hour of 
[revised manuscript text omitted]

Surface concentration comparisons presented in this study and in the supplementary material show that the volcanic $SO_2$ was observed as short singular peaks lasting from a few hours to several peaks over a short set of days. The biggest difference for the three studied plumes is for the first one during 20 to 26 September for the Manchester (GB0613A) in Great Britain and the Saint-Naizaire station (FR23181) in France, with up to a factor of four differences between simulated and measured concentrations . Both the measured and simulated concentrations during the September event were higher than the two later events, pointing to a different transport of $SO_2$ in the first event, and not only higher emissions. Higher emission fluxes are also not supported by the satellite comparison over these days either. Changes in emission flux for the EMEP/MSC-W have been shown to have an almost linear change in concentrations (not shown here); even with doubled emissions during this event the model would still simulate concentrations and burdens well below those observed. Station data presented in Schmidt et al. (2015) for these days show the same results, indicating that the models and meteorology had difficulties representing this period.

The discrepancies between the model and observations, especially for the station data show that the values presented in Table 2 contain error. Especially the model surface concentrations are low compared to observations; however the map plots show, that sometimes modelled concentrations nearby the stations reached observed levels. The area averaged concentrations presented in table 2 may therefore be close to the real concentration increase. A more thorough study of longer time series with deposition and concentration trends is needed to estimate better the increase in $SO_2$ concentrations due to the eruption at the stations.

[revised manuscript text omitted]

Studying the changes in pollution levels over Europe, $SO_X$ wet deposition showed the highest increase in the model. For the basic simulation there is 32 % more sulphate wet deposition than the model simulation with no Holuhraun emission over the 28 European Union countries, Norway and Switzerland. The regions that have the highest increase, apart from Iceland, are Northern Scandinavia and Scotland, regions that are among the least polluted in Europe. Especially the coast of Northern Norway, with a percent increase in total deposition of over 1000%, shows levels  equal to the most polluted regions in Europe Compared with observed levels  since 1980 at the Tustervatn (station in central Norway) since 1980 the 2014 model values are earlier only reached in the observations during the Grimsvötn eruption in 2011. Higher values measured at the Kårvatn station in 2014 on the coast of western Norway are due to the Holuhraun emissions. Compared to model simulations with meteorology and emission from previous years, the mean deposition levels over Norway are double that of 1990.

The difference in $SO_2$ concentrations over Europe between the no_hol and model simulations with Holuhraun emission are around 13 percent over the same 30 countries and increases occurs as short peaks in concentration levels from a few hours to some days. Due to the underestimation seen at stations during September, the uncertainty of this number is large and the increase is possibly too small. For $PM_{2.5}$ concentration, the increase is six percent., and the model shows better agreement with station observations. The biggest difference in percent increase areis seen over Scandinavia and Scotland, however these regions are among the cleanest in Europe, also with the added sulphur caused by the Holuhraun emissions. A lot of the sulphur is also deposited out over these regions by frequent precipitation. The areas that show increase in days with over 25 $\mu g/m^2$ $PM_{2.5}$ concentrations are already polluted. Even though with high emissionemissions from the volcanic fissure at Holuhraun, the increasesincrease in pollution levels are low over Europe. is 
[revised manuscript text omitted]

[Figure]

Figure 7. Daily time series of SO$_X$ deposition from The Kårvatn station in Norway. The lines represent the same as the three plots above.

[Figure]

Figure 8. Total deposition of SO$_X$ (wet and dry) over Europe from September to November for no_hol (a) and bas_hol (b) simulations and the percent increase due to the Holuhraun emissions (c). d) Shows the same as c) but zoomed into Norway and Northern Europe.

[Figure]

Figure 89. Show the same as Figure 78, but with average PM$_{2.5}$ concentration over the three months.

[Figure]

Figure 10. a) Days with exceedances of PM$_{2.5}$ over September trough November for the bas_hol model simulation. b) The increase in days from no_hol to bas_hol.

---

## Author Response (AR2)

Dear editor,

The authors would like to thank the reviewer and editor for their careful and encouraging comments.

Please find below our response to critical comments, author response are given in italics:

"I certainly think this manuscript should eventually be published, but I still feel that, at times, the writing style is too poor and too qualitative for ACP standards."

*=> We went through the whole document and revised a lot of phrases carefully. A track changed manuscript is provided as part of the response.*

"My main quibble is about their sentence saying that pollution increased by up to 50 ug/m^3 (lines 4-6 abstract) this is not true and the authors state that later themselves (lines 1-2 on page 4 of their revised paper in the attached document). They explain in the reply to my comment that they excluded these stations, but to my mind that cannot be a valid reason to state in an abstract that pollution reached up to 50 ug/m^3 because I think people agree that the pollution episodes (> 500 ug/m3) in Ireland on 6 Sep 2014 are linked to Holuhraun."

*=> We have rectified the abstract and included the comparison to the episode in Ireland into the text and graphs into the supplementary material. We had unfortunately not access to all data used by Schmidt et al 2015, and our comparison is thus more limited. It also is not a typical trans-national episode, which was interesting in the other episodes. However, we agree that this episode was very important for the characterisation of the volcanic eruption and that it must be mentioned that concentrations > 500 ug/m3 appeared.*

Considering this sentence:

"Surface observations in Europe showed concentration increases up to 50 μg/m3 averaged over an hour of SO2 from volcanic plumes passing."

the reviewer wrote the following comment: in Ireland SO2 surface concentrations increase to over 500 ug/m3 on 6 Sep 2014. I don't understand this sentence nor is it very specific. The authors should state the period of observation and location. I assume they are not talking about the volcanic pollution episode on 6 Sep 2014 as otherwise the 50 ug/m3 is too low.

*=> We have carefully checked for a correct wording around the 6 Sep episode.*

Does the terminology 'basic (bas)' in the sentence below refer to the baseline simulation?

"The control simulation is renamed basic (bas). From the observed heights, and emission fluxes given elsewhere, this simulation is the "best guess" simulation."

*=> We agree that the naming of the base/reference/bust-guess simulation is difficult. We kept the abbreviation, but added at places in the text, where we thought it useful 'best guess' to emphasize the nature of the base run.*

In the following text:

"For the first six day period, between 20 to 26 September, high concentrations of SO2 were measured over Great Brittain, and countries to the south. For the second six day period," please add 2014 behind September. The manuscript needs to be checked for such things throughout.

*=> We checked where the year 2014 was helpful to add.*

With respect to this statement:

"The detailed information will be included in the manuscript. The high SO2 concentration observed over Ireland on 6 September did not show up on many of the station that we were able to collect, so it was left out of the manuscript, but two Irish stations are shown in Figure 3."

The reviewer had the following comment:

I don't understand why this is a reason to leave out these data. Several stations in Ireland detected high SO2 pollution and backward and forward trajectory analysis shows that it is volcanic pollution.

*=> Reviewer is correct, see our comment above, we have added a paragraph to discuss the Ireland episode.*

Please rephrase the sentence below that the reviewer deem is neither correct, nor good English:

"Surface observations in Europe showed peak type concentration increases up to 50 μg/m3 averaged over an hour of SO2 concentrations from volcanic plumes passing by and lasting only for a short time."

*=> rephrased to*

*"Surface concentration comparisons presented in this study and in the supplementary material show that the volcanic $SO_2$ was observed as short singular peaks lasting a few hours or as a sequence of several peaks spread over a few days. Three episodes are picked where transnational transport is documented. "*

"The eruption ended in February 2015 and during the 6 months of eruption a total of approximately 11 (± 5) Tg SO2 may have been released (Gislason et al. 2015), and the total lava field from the fissure were 85 km2 with a volume of 1.4 km3 (vedur.is)."

Check wording, is the word area missing from this sentence?

*=> reworded to*

[revised manuscript text omitted]
 areburden is above the model value, not includingignoring the days where the OMI zoom mode minimizes the is responsible for a small coverage. The average satellite derived $SO_2$ mass burden adjusted to theassuming a 7 km reference height for satellite data are 11.17 kt $SO_2$ for satellite andis 11.2 kt, while the kernel weighted model burden in bas_hol is 8.727 kt $SO_2$ for the model.. The highest values are found at the beginning of the period, 42.111 kt $SO_2$ for the model data on 7 September, for the model, and 37.424 kt $SO_2$, on 20 September for the satellite data.. Taking into account the area in which the satellite observed $SO_2$ is found above detection limit, the satellite average column loadings are calculated to reach 70 mgm$^{-2}$ for September. Also the peaks in the middle of October, visible in Fig 2b, exhibit a satellite average column loading of 62 mgm$^{-2}$.

[revised manuscript text omitted]

---

## Author Response (AR3)

**Response to editor**

Thank you for reviewing our paper and the helpfull comments.

Author responses are given in *italics*

In the abstract, please be more clear about what you mean by a 'three months average SO2 and PM2.5 concentrations' in the following sentence:

'On three month average over Europe, SO2 and PM2.5 surface concentrations increase, due to the volcanic emissions, increased by only ten anqdand six percent over Europe, respectively.'

Do you mean that the PM2.5 concentrations increased during a particular 3-months period? If this is the case, indicate exactly the period. If not, clarify what you meant.

*Added the abreviation SON in the abstract to make more clear what period is meant*

Change 'exceedances days' to 'exceedance days' in the abstract

*Corrected it*

[revised manuscript text omitted]